# A* CCG Parsing with a Supertag and Dependency Factored Model

## Abstract

We propose a new A* CCG parsing model in which the probability of a tree is decomposed into factors of CCG categories and its dependency structure both defined on bi-directional LSTMs. Our factored model allows the precomputation of all probabilities and runs very efficiently, while modeling sentence structures explicitly via dependencies. Our model achieves the state-of-the-art results on English and Japanese CCG parsing[1].

## 1 Introduction

Supertagging in lexicalized grammar parsing is known as *almost parsing* (Bangalore and Joshi, 1999), in that each supertag is syntactically informative and most ambiguities are resolved once a correct supertag is assigned to every word. Recently this property is effectively exploited in A* Combinatory Categorial Grammar (CCG; Steedman (2000)) parsing (Lewis and Steedman, 2014; Lewis et al., 2016), in which the probability of a CCG tree $y$ on a sentence $x$ of length $N$ is the product of the probabilities of supertags (categories) $c_i$ (locally factored model):

$$P(y|x) = \prod_{i \in [1,N]} P_{tag}(c_i|x). \quad (1)$$

By not modeling every combinatory rule in a derivation, this formulation enables us to employ efficient A* search (see Section 2), which finds the most probable supertag sequence that can build a well-formed CCG tree.

Although much ambiguity is resolved with this supertagging, some ambiguity still remains. Figure 1 shows an example, where the two CCG

---

[1] We provide our software as a supplementary material.

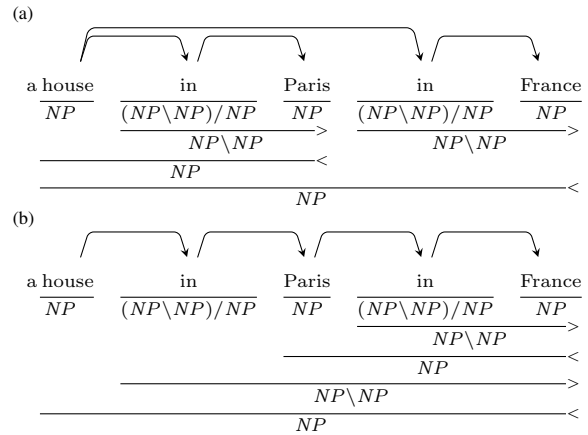

Figure 1: CCG trees that are equally likely under Eq. 1. Our model resolves this ambiguity by modeling the head of every word (dependencies).

parses are derived from the same supertags. Lewis et al.'s approach to this problem is resorting to some deterministic rule. For example, Lewis et al. (2016) employ the attach low heuristics, which is motivated by the right-branching tendency of English, and always prioritizes (b) for this type of ambiguity. Though for English it empirically works well, an obvious limitation is that it does not always derive the correct parse; consider a phrase "*a house in Paris with a garden*", for which the correct parse has the structure corresponding to (a) instead.

In this paper, we provide a way to resolve these remaining ambiguities under the locally factored model, by explicitly modeling bilexical dependencies as shown in Figure 1. Our joint model is still locally factored so that an efficient A* search can be applied. The key idea is to predict the head of every word independently as in Eq. 1 with a strong unigram model, for which we utilize the scoring model in the recent successful graph-based dependency parsing on LSTMs (Kiperwasser and Goldberg, 2016; Dozat and Manning, 2016). Specif-

ically, we extend the bi-directional LSTM (bi-LSTM) architecture of Lewis et al. (2016) predicting the supertag of a word to predict the head of it at the same time with a bilinear transformation.

The importance of modeling structures beyond supertags is demonstrated in the performance gain in Lee et al. (2016), which adds a recursive component to the model of Eq. 1. Unfortunately, this formulation loses the efficiency of the original one since it needs to compute a recursive neural network every time it searches for a new node. Our model does not resort to the recursive networks while modeling tree structures via dependencies.

We also extend the tri-training method of Lewis et al. (2016) to learn our model with dependencies from unlabeled data. On English CCGbank test data, our model with this technique achieves 88.8% and 94.0% in terms of labeled and unlabeled F1, which mark the best scores so far.

Besides English, we provide experiments on Japanese CCG parsing. Japanese employs freer word order dominated by the case markers and a deterministic rule such as the attach low method may not work well. We show that this is actually the case; our method outperforms the simple application of Lewis et al. (2016) in a large margin, 10.0 points in terms of clause dependency accuracy.

## 2 Background

Our approach is build on A* CCG parsing (Section 2.1), which we extend in Section 3 with a head prediction model on bi-LSTMs (Section 2.2).

### 2.1 Supertag-factored A* CCG Parsing

CCG has a nice property that since every category is highly informative about attachment decisions, assigning it to every word (*supertagging*) determines most of its syntactic structure. Lewis and Steedman (2014) utilize this characteristics of the grammar. Let a CCG tree $\boldsymbol{y}$ be a list of categories $\langle c_1, \ldots, c_N \rangle$ and a derivation on it. Their model looks for the most probable $\boldsymbol{y}$ given a sentence $\boldsymbol{x}$ from the set $Y(\boldsymbol{x})$ of possible CCG trees under the model of Eq. 1:

$$\hat{\boldsymbol{y}} = \arg\max_{\boldsymbol{y} \in Y(\boldsymbol{x})} \sum_{i \in [1,N]} \log P_{tag}(c_i | \boldsymbol{x}).$$

Since this score is factored into each supertag, they call it *supertag-factored*.

Exact inference of this problem is possible by A* parsing (Klein and D. Manning, 2003), which uses the following two scores on a chart:

$$
\begin{aligned}
b(C_{i,j}) &= \sum_{c_k \in \boldsymbol{c}_{i,j}} \log P_{tag}(c_k | \boldsymbol{x}), \\
a(C_{i,j}) &= \sum_{k \in [1,N] \setminus [i,j]} \max_{c_k} \log P_{tag}(c_k | \boldsymbol{x}),
\end{aligned}
$$

where $C_{i,j}$ is a chart item called an *edge*, which abstracts parses spanning interval $[i, j]$ rooted by category $C$. The chart maps each edge to the derivation with the highest score, i.e., the Viterbi parse for $C_{i,j}$. $\boldsymbol{c}_{i,j}$ is the sequence of categories on such Viterbi parse, and thus $b$ is called the Viterbi inside score, while $a$ is the approximation (upper bound) of the Viterbi outside score.

A* parsing is a kind of CKY chart parsing augmented with an agenda, a priority queue that keeps the edges to be explored. At every step it pops the edge $e$ with the highest priority $b(e) + a(e)$ and inserts that into the chart, and enqueue any edges that can be built by combining $e$ with other edges in the chart. The algorithm terminates when an edge $C_{1,N}$ is popped from the agenda.

A* search for this model is quite efficient because both $b$ and $a$ can be obtained from the unigram category distribution on every word, which can be precomputed before search. The heuristics $a$ gives an upperbound on the true Viterbi outside score (i.e., admissible). Along with this the condition that the inside score never decreases by expansion (monotonicity) guarantees that the first found derivation on $C_{1,N}$ is always optimal. $a(C_{i,j})$ matches the true outside score if the one-best category assignments on the outside words ($\arg\max_{c_k} \log P_{tag}(c_k | \boldsymbol{x})$) can comprise a well-formed tree with $C_{i,j}$, which is generally not true.

**Scoring model** For modeling $P_{tag}$, Lewis and Steedman (2014) use a log-linear model with features from a fixed window context. Lewis et al. (2016) extend this with bi-LSTMs, which encode the complete sentence and capture the long range syntactic information. We base our model on this bi-LSTM architecture, and extend it to modeling a head word at the same time.

**Attachment ambiguity** In A* search, an edge with the highest priority $b + a$ is searched first, but as shown in Figure 1 the same categories (with the same priority) may sometimes derive different

trees. In Lewis and Steedman (2014), they prioritize the parse with longer dependencies, which they judge with a conversion rule from a CCG tree to a dependency tree (Section 4). Lewis et al. (2016) employ another heuristics prioritizing low attachments of constituencies, but inevitably these heuristics cannot be flawless in any situations. We provide a simple solution to this problem by explicitly modeing bilexical dependencies.

### 2.2 Bi-LSTM Dependency Parsing

For modeling dependencies, we borrow the idea from the recent graph-based neural dependency parsing (Kiperwasser and Goldberg, 2016; Dozat and Manning, 2016) in which each dependency arc is scored directly on the outputs of bi-LSTMs. Though the model is first-order, bi-LSTMs enable conditioning on the entire sentence and lead to the state-of-the-art performance. Note that this mechanism is similar to modeling of the supertag distribution discussed above, in that for each word the distribution of the head choice is unigram and can be precomputed. As we will see this keeps our joint model still locally factored and A* search tractable. For score calculation, we use an extended bilinear transformation by Dozat and Manning (2016) that models the bias term as well, which they call *biaffine*.

## 3 Proposed Method

### 3.1 A* parsing with Supertag and Dependency Factored Model

We define a CCG tree $\boldsymbol{y}$ for a sentence $\boldsymbol{x} = \langle x_i, \ldots, x_N \rangle$ as a triplet of a list of CCG categories $\boldsymbol{c} = \langle c_1, \ldots, c_N \rangle$, dependencies $\boldsymbol{h} = \langle h_1, \ldots, h_N \rangle$, and the derivation, where $h_i$ is the head index of $x_i$. Our model is defined as follows:

$$P(\boldsymbol{y}|\boldsymbol{x}) = \prod_{i \in [1,N]} P_{tag}(c_i|\boldsymbol{x}) \prod_{i \in [1,N]} P_{dep}(h_i|\boldsymbol{x}). \tag{2}$$

The added term $P_{dep}$ is a unigram distribution of the head choice.

A* search is still tractable under this model. The search problem is changed as:

$$\hat{\boldsymbol{y}} = \arg\max_{\boldsymbol{y} \in Y(\boldsymbol{x})} \left( \sum_{i \in [1,N]} \log P_{tag}(c_i|\boldsymbol{x}) \right.$$
$$\left. + \sum_{i \in [1,N]} \log P_{dep}(h_i|\boldsymbol{x}) \right),$$

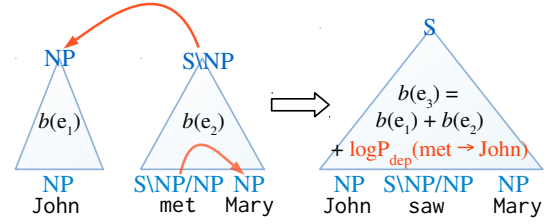

Figure 2: Viterbi inside score for edge $e_3$ under our model is the sum of those of $e_1$ and $e_2$ and the score of dependency arc going from the head of $e_2$ to that of $e_1$ (the head direction changes according to the child categories).

and the inside score is given by:

$$b(C_{i,j}) = \sum_{c_k \in \boldsymbol{c}_{i,j}} \log P_{tag}(c_k|\boldsymbol{x}) \tag{3}$$
$$+ \sum_{k \in [i,j] \setminus \{root(\boldsymbol{h}_{i,j}^C)\}} \log P_{dep}(h_k|\boldsymbol{x}),$$

where $\boldsymbol{h}_{i,j}^C$ is a dependency subtree for the Viterbi parse on $C_{i,j}$ and $root(\boldsymbol{h})$ returns the root index. We exclude the head score for the subtree root token since it cannot be resolved inside $[i,j]$. This causes the mismatch between the goal inside score $b(C_{1,N})$ and the true model score (log of Eq. 2), which we adjust by adding a special unary rule that is always applied to the popped goal edge $C_{1,N}$.

We can calculate the dependency terms in Eq. 3 on the fly when expanding the chart. Let the current popped edge be $A_{i,k}$, which will be combined with $B_{k,j}$ into $C_{i,j}$. The key observation is that only one dependency arc (between $root(\boldsymbol{h}_{i,k}^A)$ and $root(\boldsymbol{h}_{k,j}^B)$) is resolved at every combination (see Figure 2). For every rule $C \to A\ B$ we can define the head direction (see Section 4) and $P_{dep}$ is obtained accordingly. For example, when the right child $B$ becomes the head, $b(C_{i,j}) = b(A_{i,k}) + b(B_{k,j}) + \log P_{dep}(h_l = m|\boldsymbol{x})$, where $l = root(\boldsymbol{h}_{i,k}^A)$ and $m = root(\boldsymbol{h}_{k,j}^B)$ ($l < m$).

The Viterbi outside score is changed as:

$$a(C_{i,j}) = \sum_{k \in [1,N] \setminus [i,j]} \max_{c_k} \log P_{tag}(c_k|\boldsymbol{x})$$
$$+ \sum_{k \in L} \max_{h_k} \log P_{dep}(h_k|\boldsymbol{x}),$$

where $L = [1,N] \setminus [k'|k' \in [i,j], root(\boldsymbol{h}_{i,j}^C) \neq k']$. We regard $root(\boldsymbol{h}_{i,j}^C)$ as an outside word since its head is undefined yet. For every outside word we independently assign the weight of its argmax

head, which may not comprise a well-formed dependency tree. We initialize the agenda by adding an item for every supertag $C$ and word $x_i$ with the score $a(C_{i,i}) = \sum_{k \in I \setminus \{i\}} \max \log P_{tag}(c_k|\boldsymbol{x}) + \sum_{k \in I} \max \log P_{dep}(h_k|\boldsymbol{x})$. Note that the dependency component of it is the same for every word.

### 3.2 Network Architecture

Following Lewis et al. (2016) and Dozat and Manning (2016), we model $P_{tag}$ and $P_{dep}$ using bi-LSTMs for exploiting the entire sentence to capture the long range phenomena. See Figure 3 for the overall network architecture, where $P_{tag}$ and $P_{dep}$ share the common bi-LSTM hidden vectors.

First we map every word $x_i$ to their hidden vector $\boldsymbol{r}_i$ with bi-LSTMs. The input to the LSTMs is word embeddings, which we describe in Section 6. We add special start and end tokens to each sentence with the trainable parameters following Lewis et al. (2016). For $P_{dep}$, we use the biaffine transformation in Dozat and Manning (2016):

$$\boldsymbol{g}_i^{dep} = MLP_{child}^{dep}(\boldsymbol{r}_i),$$
$$\boldsymbol{g}_{h_i}^{dep} = MLP_{head}^{dep}(\boldsymbol{r}_{h_i}),$$
$$P_{dep}(h_i|\boldsymbol{x}) \tag{4}$$
$$\propto \exp((\boldsymbol{g}_i^{dep})^{\mathrm{T}} W_{dep} \boldsymbol{g}_{h_i}^{dep} + \boldsymbol{w}_{dep} \boldsymbol{g}_{h_i}^{dep}),$$

where $MLP$ is a multilayered perceptron. Though Lewis et al. (2016) simply use a MLP for mapping $\boldsymbol{r}_i$ to $P_{tag}$, we additionally utilize the hidden vector of the most probable head $h_i = \arg\max_{h_i'} P_{dep}(h_i'|\boldsymbol{x})$, and apply $\boldsymbol{r}_i$ and $\boldsymbol{r}_{h_i}$ to a bilinear function:[2]

$$\boldsymbol{g}_i^{tag} = MLP_{child}^{tag}(\boldsymbol{r}_i),$$
$$\boldsymbol{g}_{h_i}^{tag} = MLP_{head}^{tag}(\boldsymbol{r}_{h_i}), \tag{5}$$
$$\boldsymbol{\ell} = (\boldsymbol{g}_i^{tag})^{\mathrm{T}} \boldsymbol{U}_{tag} \boldsymbol{g}_{h_i}^{tag} + W_{tag} \begin{bmatrix} \boldsymbol{g}_i^{tag} \\ \boldsymbol{g}_{h_i}^{tag} \end{bmatrix} + \boldsymbol{b}_{tag},$$
$$P_{tag}(c_i|\boldsymbol{x}) \propto \exp(\boldsymbol{\ell}_c),$$

where $\boldsymbol{U}_{tag}$ is a third order tensor. As in Lewis et al. these values can be precomputed before search, which makes our A* parsing quite efficient.

### 4 CCG to Dependency Conversion

Now we describe our conversion rules from a CCG tree to a dependency one, which we use in two purposes: 1) creation of the training data for the de-

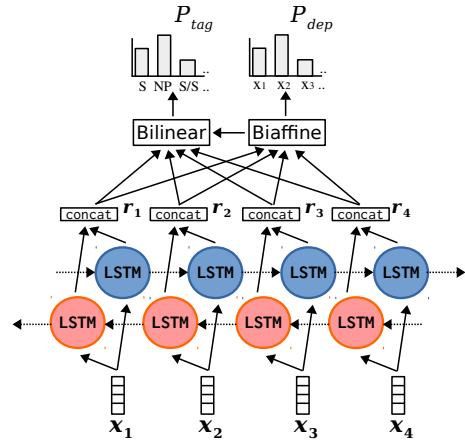

Figure 3: Neural networks of our supertag and dependency factored model. First we map every word $x_i$ to a hidden vector $\boldsymbol{r}_i$ by bi-LSTMs, and then apply biaffine (Eq. 4) and bilinear (Eq. 5) transformations to obtain the distributions of dependency heads ($P_{dep}$) and supertags ($P_{tag}$).

pendency component of our model; and 2) extraction of a dependency arc at each combinatory rule during A* search (Section 3.1). Lewis and Steedman (2014) describe one way to extract dependencies from the CCG tree (LEWISRULE). Below in addition to this we describe two simpler alternatives (HEADFIRST and HEADFINAL), and see the effects on parsing performance in our experiments (Section 6). See Figure 4 for the overview.

**LEWISRULE** This is the same as the conversion rule in Lewis and Steedman (2014). As shown in Figure 4c the output looks a familiar English dependency tree.

For forward application and (generalized) forward composition, we define the head to be the left argument of the combinatory rule, unless it matches either $X/X$ or $X/(X \setminus Y)$, in which case the right argument is the head. For example, on *"Black Monday"* in Figure 4a we choose *Monday* as the head of *Black*. For the backward rules, the conversions are defined as the reverse of the corresponding forward rules. For other rules, *RemovePunctuation* (*rp*) chooses the non punctuation argument as a head, while *Conjunction* ($\Phi$) chooses the right argument.[3]

One issue when applying this method for ob-

---

[2] This is inspired by the formulation of label prediction in Dozat and Manning (2016).

[3] When applying LEWISRULE to Japanese, we ignore the identity of the feature values in determining the head argument; In "tabe ta" (eat *PAST*), the category of auxiliary verb "ta" is $S_{f_1} \setminus S_{f_2}$ with $f_1 \neq f_2$, thus $S_{f_1} \neq S_{f_2}$. Though it is not $X \setminus X$, we define "ta" is headed by "tabe", as removing the feature values, it matches $X \setminus X$.

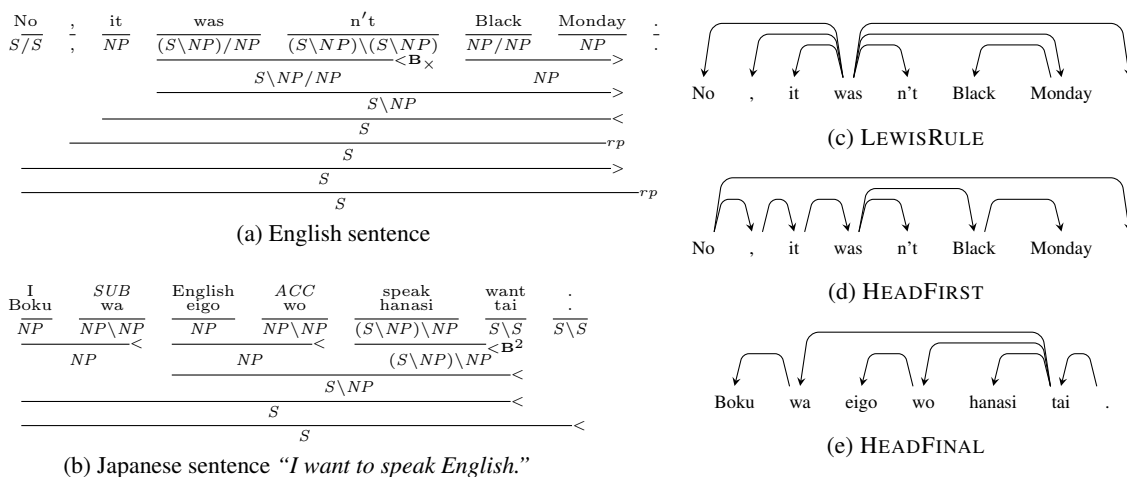

Figure 4: Examples of applying conversion rules in Section 4 to English and Japanese sentences.

taining the training data is that due to the mismatch between the rule set of our CCG parser, for which we follow Lewis and Steedman (2014), and the grammar in English CCGbank (Hockenmaier and Steedman, 2007) we cannot extract dependencies from some of annotated CCG trees.[4] For this reason, we instead obtain the training data for this method from the original dependency annotations on CCGbank. Forutnatelly the dependency annotations of CCGbank matches LEWISRULE above in most cases and thus they can be a good approximation to it.

**HEADFINAL**   Among SOV languages, Japanese is known as a strictly head final language, meaning that the head of every word always follows it. Japanese dependency parsing (Uchimoto et al., 1999; Kudo and Matsumoto, 2002) has exploited this property explicitly by only allowing left-to-right dependency arcs. Inspired by this tradition, we try a simple HEADFINAL rule in Japanese CCG parsing, in which we always select the right argument as a head. For example we obtain the head final dependency tree in Figure 4e from the Japanese CCG tree in Figure 4b.

**HEADFIRST**   We apply the similar idea as HEADFINAL into English. Since English has the opposite, SVO word order, we define the simple "head first" rule, in which the left argument always becomes the head (Figure 4d).

Though this conversion may look odd at first sight it also has some advantages over LEWIS-

RULE. First, since the model with LEWISRULE is trained on the CCGbank dependencies, at inference, occasionally the two components $P_{dep}$ and $P_{tag}$ cause some conflicts on their predictions. For example, the true Viterbi parse may have a lower score in terms of dependencies, in which case the parser slows down and may degrade the accuracy. HEADFIRST, in contract, does not suffer from such conflicts. Second, by forcing the direction of arcs, the prediction of heads becomes easier, meaning that the dependency predictions become more reliable. Later we show this is in fact the case for existing dependency parsers (see Section 5), and in practice, we find that this simple conversion rule leads to the higher parsing scores than LEWISRULE on English (Section 6).

## 5   Tri-training

We extend the existing tri-training method to our models and apply it to our English parsers.

Tri-training is one of the semi-supervised methods, in which the outputs of two parsers on unlabeled data are intersected to create (silver) new training data. This method is successfully applied to dependency parsing (Weiss et al., 2015) and CCG supertagging (Lewis et al., 2016).

We simply combine the two previous approaches. Lewis et al. (2016) obtain their silver data annotated with the high quality supertags. Since they make this data publicly available [5], we obtain our silver data by assigning dependency structures on top of them.[6]

---

[4] For example, the combinatory rules in Lewis and Steedman (2014) do not contain $N_{conj} \rightarrow N\ N$ in CCGbank. Another difficulty is that in English CCGbank a combinatory rule name is not annotated explicitly.

[5] https://github.com/uwnlp/taggerflow

[6] We annotate POS tags on this data using Stanford POS tagger (Toutanova et al., 2003).

We train two very different dependency parsers from the training data extracted from CCGbank Section 02-21. This training data differs depending on our dependency conversion strategy (Section 4). For LEWISRULE, we extract the original dependency annotations of CCGbank. For HEADFIRST, we extract the head first dependencies from the CCG trees. Note that we cannot annotate dependency labels so we assign a dummy "none" label to every arc. The first parser is graph-based `RBGParser` (Lei et al., 2014) with the default settings except that we train an unlabeled parser and use word embeddings of Turian et al. (2010). The second parser is transition-based `lstm-parser` (Dyer et al., 2015) with the default parameters.

On the development set (Section 00), with LEWISRULE dependencies `RBGParser` shows 93.8% unlabeled attachment score while that of `lstm-parser` is 92.5% using gold POS tags. Interestingly, the parsers with HEADFIRST dependencies achive higher scores: 94.9% by `RBGParser` and 94.6% by `lstm-parser`, suggesting that HEADFIRST dependencies are more parsable. For both dependencies, we obtain more than 1.7 million sentences on which two parsers agree.

Following Lewis et al. (2016), we include 15 copies of CCGbank training set when using these silver data. Also to make effects of the tritrain samples smaller we multiply their loss by 0.4.

# 6 Experiments

We perform experiments on English and Japanese CCGbanks.

## 6.1 English Experimental Settings

We follow the standard data splits and use Sections 02-21 for training, Section 00 for development, and Section 23 for final evaluation. We report labeled and unlabeled F1 of the extracted CCG semantic dependencies obtained using `generate` program supplied with `C&C` parser.

For our models, we adopt the pruning strategies in Lewis and Steedman (2014) and allow at most 50 categories per word, use a variable-width beam with $\beta = 0.00001$, and utilize a tag dictionary, which maps frequent words to the possible supertags[7]. Unless otherwise stated, we only al-

low normal form parses (Eisner, 1996; Hockenmaier and Bisk, 2010), choosing the same subset of the constaints as Lewis and Steedman (2014).

We use as word representation the concatenatination of word vectors initialized to GloVe[8] (Pennington et al., 2014), and randomly initialized prefix and suffix vectors of the length 1 to 4, which is inspired by Lewis et al. (2016). All affixes appearing less than two times in the training data are mapped to "UNK".

Other model configurations are: 4-layer bi-LSTMs with left and right 300-dimensional LSTMs, 1-layer 100-dimensional MLPs with ELU non-linearity (Clevert et al., 2015) for all $MLP_{child}^{dep}$, $MLP_{head}^{dep}$, $MLP_{child}^{tag}$ and $MLP_{head}^{tag}$, and the Adam optimizer with $\beta_1 = 0.9, \beta_2 = 0.9$, L2 norm ($1e^{-6}$), and lerning rate decay with the ratio 0.75 for every 2,500 iteration starting from $2e^{-3}$, which is shown to be effective for training the biaffine parser (Dozat and Manning, 2016).

## 6.2 Japanese Experimental Settings

We follow the default train/dev/test splits of Japanese CCGbank (Uematsu et al., 2013). For the baselines, we use an existing shift-reduce CCG parser implemented in an NLP tool Jigg[9] (Noji and Miyao, 2016), and our implementation of the supertag-factored model using bi-LSTMs.

For Japanese, we use as word representation the concatenatination of word vectors initialized to Japanese Wikipedia Entity Vector[10], and 100-dimensional vectors computed from randomly initialized 50-dimensional character embeddings through convolution (dos Santos and Zadrozny, 2014). We do not use affix vectors as affixes are less informative in Japanese. All characters appearing less than two times are mapped to "UNK". We use the same parameter settings as English for bi-LSTMs, MLPs, and optimization.

One issue in Japanese experiments is evaluation. The Japanese CCGbank is encoded in a different format than the English bank, and no standalone script for extracting semantic dependencies is available yet. For this reason, we evaluate the parser outputs by converting them to *bunsetsu dependencies*, the syntactic representation ordinary used in Japanese NLP (Kudo and Matsumoto, 2002). Given a CCG tree, we obtain this by first

---

[7]We use the same tag dictionary provided with their bi-LSTM model.

[8]http://nlp.stanford.edu/projects/glove/

[9]https://github.com/mynlp/jigg

[10]http://www.cl.ecei.tohoku.ac.jp/~m-suzuki/jawiki_vector/

| Method | Labeled | Unlabeled |
|---|---|---|
| *CCGbank* | | |
| LEWISRULE w/o dep | 85.8 | 91.7 |
| LEWISRULE | 86.0 | 92.5 |
| HEADFIRST w/o dep | 85.6 | 91.6 |
| HEADFIRST | **86.6** | **92.8** |
| *Tri-training* | | |
| LEWISRULE | 86.9 | 93.0 |
| HEADFIRST | **87.6** | **93.3** |

Table 1: Parsing results (F1) on English development set. "w/o dep" means that the model discards dependency components at prediction.

| Method | Labeled | Unlabeled | # violations |
|---|---|---|---|
| *CCGbank* | | | |
| LEWISRULE w/o dep | 85.8 | 91.7 | 2732 |
| LEWISRULE | 85.4 | 92.2 | 283 |
| HEADFIRST w/o dep | 85.6 | 91.6 | 2773 |
| HEADFIRST | **86.8** | **93.0** | **89** |
| *Tri-training* | | | |
| LEWISRULE | 86.7 | 92.8 | 253 |
| HEADFIRST | **87.7** | **93.5** | **66** |

Table 2: Parsing results (F1) on English development set when excluding the normal form constraints. # violations is the number of combinations violating the constraints on the outputs.

segment a sentence into bunsetsu (chunks) using CaboCha[11] and extract dependencies that cross a bunsetsu boundary after obtaining the word-level, head final dependencies as in Figure 4b. For example, the sentence in Figure 4e is segmented as "*Boku wa | eigo wo | hanashi tai*", from which we extract two dependencies (*Boku wa*) ← (*hanashi tai*) and (*eigo wo*) ← (*hanashi tai*). We perform this conversion for both gold and output CCG trees and calculate the (unlabeled) attachment accuracy. Though this is imperfect, it can detect important parse errors such as attachment errors and thus can be a good proxy for the performance as a CCG parser.

### 6.3 English Parsing Results

**Effect of Dependency** We first see how the dependency components added in our model affect the performance. Table 1 shows the results on the development set with the several configurations, in which "w/o dep" means discarding the dependency terms of the model and applying the attach low heuristics (Section 1) instead (i.e., a supertag-factored model; Section 2.1). We can see that for

---

[11]http://taku910.github.io/cabocha/

| Method | Labeled | Unlabeled |
|---|---|---|
| *CCGbank* | | |
| C&C (Clark and R. Curran, 2007) | 85.5 | 91.7 |
| w/ LSTMs (Vaswani et al., 2016) | **88.3** | - |
| EasySRL (Lewis et al., 2016) | 87.2 | - |
| EasySRL_reimpl | 86.8 | 92.3 |
| HEADFIRST w/o NF (Ours) | 87.7 | **93.4** |
| *Tri-training* | | |
| EasySRL (Lewis et al., 2016) | 88.0 | 92.9 |
| neuralccg (Lee et al., 2016) | 88.7 | 93.7 |
| HEADFIRST w/o NF (Ours) | **88.8** | **94.0** |

Table 3: Parsing results (F1) on English test set (Section 23).

both LEWISRULE and HEADFIRST, adding dependency terms improves the performance.

**Choice of Dependency Conversion Rule** To our surprise, our simple HEADFIRST strategy always leads to better results than the linguistically motivated LEWISRULE. The absolute improvements by tri-training are equally large (about 1.0 points), suggesting that our model with dependencies can also be benefited from the silver data.

**Excluding Normal Form Constraints** One advantage of HEADFIRST is that the direction of arcs is always right, making the structures simpler and more parsable (Section 5). From another viewpoint, this fixed direction means that the constituent structure behind a (head first) dependency tree is unique. Since the constituent structures of CCGbank trees basiclly follow the normal form (NF), we hypothesize that the model learned with HEADFIRST has an ability to force the outputs in NF automatically. We summarize the results without the NF constraints in Table 2, which shows that the above argument is correct; the number of violating NF rules on the outputs of HEADFIRST is much smaller than that of LEWISRULE (89 vs. 283). Interestingly the scores of HEADFIRST slighly increase from the models with NF (e.g., 86.8 vs. 86.6 for CCGbank), suggesting that the NF constraints hinder the search of HEADFIRST models occasionally.

**Results on Test Set** Parsing results on the test set (Section 23) are shown in Table 3, where we compare our best-performance HEADFIRST dependency model without NF constraints with the several existing parsers. In CCGbank experiment, our parser shows the better result than all the baseline parsers except C&C with LSTM supertagger (Vaswani et al., 2016). Our parser outper-

|            | neuralccg | EasySRL_reimpl | Ours |
|------------|-----------|----------------|------|
| *Tagging*  | **21.7**  | 14.0           | 8.6  |
| *A\* Search* | 16.7    | **200.6**      | 89.1 |
| *Total*    | 9.33      | **13.1**       | 7.9  |

Table 4: Results of the efficiency experiment, where each number is the number of sentences processed per second. We compare our proposed parser against `neuralccg` and our reimplementation of `EasySRL`.

forms `EasySRL` by 0.5% and our reimplementation of that parser (`EasySRL_reimpl`) by 0.9% in terms of labeled F1. In tri-training experiment, our parser shows much increased performance of 88.8% labeled F1 and 94.0% unlabeled F1, outperforming the current state-of-the-art `neuralccg` (Lee et al., 2016) that uses recursive neural networks by 0.1 point and 0.3 point in terms of labeled and unlabeled F1. This is the best reported F1 in English CCG parsing.

**Efficiency Comparison** We compare the efficiency of our parser with `neuralccg` and `EasySRL_reimpl`.[12] The results are shown in Table 4. Our parser lags behind in the overall parsing speed to `neuralccg` and `EasySRL_reimpl`. When we go into the details, our supertagger runs slower than those of `neuralccg` and `EasySRL_reimpl`, while in A\* search, our parser processes over 5 times more sentences than `neuralccg`. The delay in supertagging can be attributed to several factors, in particular, the implementation (Python vs. C++) and the number of parameters, especially the number of bi-LSTM layers (4 vs. 2). We note that there are many implementation differences in our parsers (C++ A\* parser with neural network model implemented with Chainer (Tokui et al., 2015)) and `neuralccg` (Java parser with C++ TensorFlow (Abadi et al., 2015) supertagger and recursive neural model in C++ DyNet (Neubig et al., 2017))[13].

### 6.4 Japanese Parsing Result

We show the results of the Japanese parsing experiment in Table 5. The simple application of Lewis

| Method | Category | Bunsetsu Dep. |
|--------|----------|---------------|
| Noji and Miyao (2016) | 93.0 | 87.5 |
| Supertag model | 93.7 | 81.5 |
| LEWISRULE (Ours) | 93.8 | 90.8 |
| HEADFINAL (Ours) | **94.1** | **91.5** |

Table 5: Results of Japanese CCGbank.

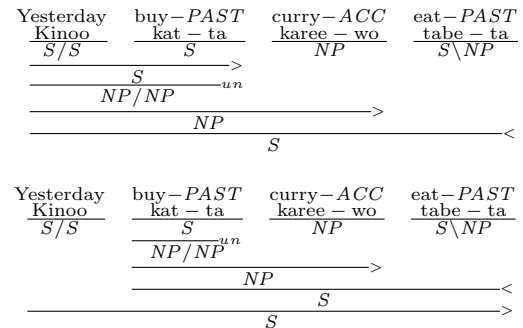

Figure 5: Examples of ambiguous Japanese sentence given fixed supertags. The English translation is *"I ate the curry I bought yesterday"*.

et al. (2016) (Supertag model) is not effective for Japanese, showing the lowest attachment score of 81.5%. We observe a performance boost with our method, especially with HEADFINAL dependencies, which outperforms the baseline shift-reduce parser by 1.1 points on category assignments and 4.0 points on attachments.

The degraded results of the simple application of the supertag-factored model can be attributed to the fact that the structure of a Japanese sentence is still highly ambiguous given the supertags (Figure 5). This is particularly the case in constructions where phrasal adverbial/adnominal modifiers (with the supertag $S/S$) are involved. The result suggests the importance of modeling dependencies in some languages, at least Japanese.

## 7 Conclusion

In this work, we have proposed a new CCG A\* parsing method, in which the probability of a CCG tree is decomposed into local factors of the CCG categories and its dependency structure. By explicitly modeling the dependency structure, we do not require any deterministic heuristics to resolve attachment ambiguities, and keep the model factored so that all the probabilities can be precomputed before running the search. Our parser efficiently finds the the optimal parses, achieving the state-of-the-art performance in both English and Japanese parsing.

---

[12]This experiment is performed on a laptop with 4-thread 2.0 GHz CPU.

[13] There seems to be a room for optimizing our parser's efficiency. We found that our `EasySRL_reimpl` is slower than the original implementation of the supertag-factored model by Lewis et al. (2016), which `neuralccg` uses internally.

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
