# Peer review of "A* CCG Parsing with a Supertag and Dependency Factored Model"

_ACL 2017 — decision unknown_

[Official Review · Reviewer 1 · rating 4 · confidence 5]
soundness 5 · originality 5 · clarity 4 · impact 3 · substance 4 · appropriateness 5 · meaningful comparison 3 · presentation format Oral Presentation

- Strengths:
This paper presents an extension to A* CCG parsing to include dependency
information.  Achieving this while maintaining speed and tractability is a very
impressive feature of this approach.  The ability to precompute attachments is
a nice trick.                  I also really appreciated the evaluation of the
effect of
the
head-rules on normal-form violations and would love to see more details on the
remaining cases.

- Weaknesses:
I'd like to see more analysis of certain dependency structures.  I'm
particularly interested in how coordination and relative clauses are handled
when the predicate argument structure of CCG is at odds with the dependency
structures normally used by other dependency parsers.

- General Discussion:
I'm very happy with this work and feel it's a very nice contribution to the
literature.  The only thing missing for me is a more in-depth analysis of the
types of constructions which saw the most improvement (English and Japanese)
and a discussion (mentioned above) reconciling Pred-Arg dependencies with those
of other parsers.

[Official Review · Reviewer 2 · rating 4 · confidence 5]
soundness 5 · originality 5 · clarity 4 · impact 3 · substance 4 · appropriateness 5 · meaningful comparison 3 · presentation format Oral Presentation

This paper describes a state-of-the-art CCG parsing model that decomposes into
tagging and dependency scores, and has an efficient A* decoding algorithm.
Interestingly, the paper slightly outperforms Lee et al. (2016)'s more
expressive global parsing model, presumably because this factorization makes
learning easier. It's great that they also report results on another language,
showing large improvements over existing work on Japanese CCG parsing. One
surprising original result is that modeling the first word of a constituent as
the head substantially outperforms linguistically motivated head rules. 

Overall this is a good paper that makes a nice contribution. I only have a few
suggestions:
- I liked the way that the dependency and supertagging models interact, but it
would be good to include baseline results for simpler variations (e.g. not
conditioning the tag on the head dependency).
- The paper achieves new state-of-the-art results on Japanese by a large
margin. However, there has been a lot less work on this data - would it also be
possible to train the Lee et al. parser on this data for comparison?
- Lewis, He and Zettlemoyer (2015) explore combined dependency and supertagging
models for CCG and SRL, and may be worth citing.